# Deep Structured Prediction with Nonlinear Output Transformations

**Colin Graber**
cgraber2@illinois.edu

**Ofer Meshi**[†]
meshi@google.com

**Alexander Schwing**
aschwing@illinois.edu

University of Illinois at Urbana-Champaign
[†]Google

## Abstract

Deep structured models are widely used for tasks like semantic segmentation, where explicit correlations between variables provide important prior information which generally helps to reduce the data needs of deep nets. However, current deep structured models are restricted by oftentimes very local neighborhood structure, which cannot be increased for computational complexity reasons, and by the fact that the output configuration, or a representation thereof, cannot be transformed further. Very recent approaches which address those issues include graphical model inference *inside* deep nets so as to permit subsequent non-linear output space transformations. However, optimization of those formulations is challenging and not well understood. Here, we develop a novel model which generalizes existing approaches, such as structured prediction energy networks, and discuss a formulation which maintains applicability of existing inference techniques.

## 1 Introduction

Nowadays, machine learning models are used widely across disciplines from computer vision and natural language processing to computational biology and physical sciences. This wide usage is fueled, particularly in recent years, by easily accessible software packages and computational resources, large datasets, a problem formulation which is general enough to capture many cases of interest, and, importantly, trainable high-capacity models, *i.e.*, deep nets.

While deep nets are a very convenient tool these days, enabling rapid progress in both industry and academia, their training is known to require significant amounts of data. One possible reason is the fact that prior information on the structural properties of output variables is not modeled explicitly. For instance, in semantic segmentation, neighboring pixels are semantically similar, or in disparity map estimation, neighboring pixels often have similar depth. The hope is that if such structural assumptions hold true in the data, then learning becomes easier (*e.g.*, smaller sample complexity) [10]. To address a similar shortcoming of linear models, in the early 2000's, structured models were proposed to augment support vector machines (SVMs) and logistic regression. Those structured models are commonly referred to as 'Structured SVMs' [52, 54] and 'conditional random fields' [27] respectively.

More recently, structured models have also been combined with deep nets, first in a two-step training setup where the deep net is trained before being combined with a structured model, *e.g.*, [1, 8], and then by considering a joint formulation [53, 59, 9, 42]. In these cases, structured prediction is used *on top* of a deep net, using simple models for the interactions between output variables, such as plain summation. This formulation may be limiting in the type of interactions it can capture. To address this shortcoming, very recently, efforts have been conducted to include structured prediction *inside*, *i.e.*, not on top of, a deep net. For instance, *structured prediction energy networks* (SPENs) [3, 4]

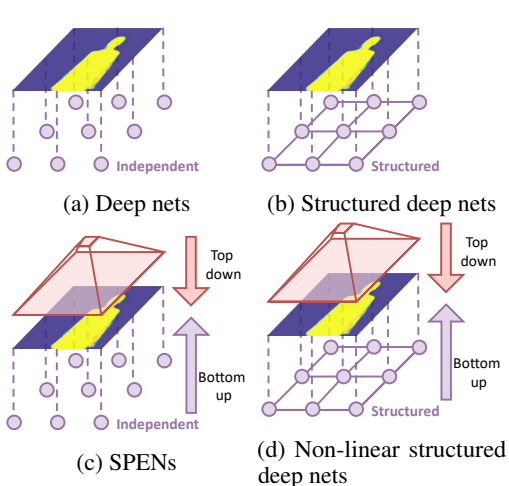

(a) Deep nets

(b) Structured deep nets

(c) SPENs

(d) Non-linear structured deep nets

Figure 1: Comparison between different model types.

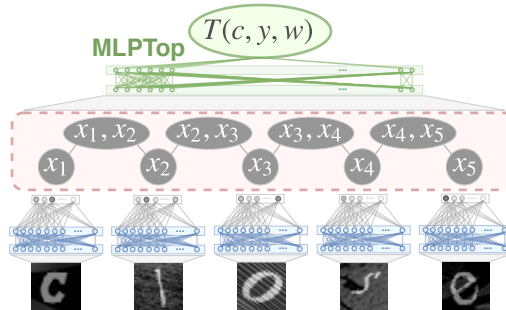

Figure 2: A diagram of the proposed nonlinear structured deep network model. Each image is transformed via a 2-layer MLP ($H$) into a 26-dimensional feature representation. Structured inference uses this representation to provide a feature vector $y$ which is subsequently transformed by another 2-layer MLP ($T$) to produce the final model score.

were proposed to reduce the excessively strict inductive bias that is assumed when computing a score vector with one entry per output space configuration. Different from the aforementioned classical techniques, SPENs compute independent prediction scores for each individual component of the output as well as a global score which is obtained by passing a complete output prediction through a deep net. Unlike prior approaches, SPENs do not allow for the explicit specification of output structure, and structural constraints are not maintained during inference.

In this work, we represent output variables as an intermediate structured layer in the middle of the neural architecture. This gives the model the power to capture complex nonlinear interactions between output variables, which prior deep structured methods do not capture. Simultaneously, structural constraints are enforced during inference, which is not the case with SPENs. We provide two intuitive interpretations for including structured prediction inside a deep net rather than at its output. First, this formulation allows one to explicitly model local output structure while simultaneously assessing global output coherence in an implicit manner. This increases the expressivity of the model without incurring the cost of including higher-order potentials within the explicit structure. A second view interprets learning of the network above the structured 'output' as training of a loss function which is suitable for the considered task.

Including structure inside deep nets isn't trivial. For example, it is reported that SPENs are hard to optimize [4]. To address this issue, here, we discuss a rigorous formulation for structure inside deep nets. Different from SPENs which apply a continuous relaxation to the output space, here, we use a Lagrangian framework. One advantage of the resulting objective is that any classical technique for optimization over the structured space can be readily applied.

We demonstrate the effectiveness of our proposed approach on real-world applications, including OCR, image tagging, multilabel classification and semantic segmentation. In each case, the proposed approach is able to improve task performance over deep structured baselines.

## 2 Related Work

We briefly review related work and contrast existing approaches to our formulation.

**Structured Prediction:** Interest in structured prediction sparked from the seminal works of Lafferty et al. [27], Taskar et al. [52], Tsochantaridis et al. [54] and has continued to grow in recent years. These techniques were originally formulated to augment linear classifiers such as SVMs or logistic regression with a model for correlations between multiple variables of interest. Although the prediction problem (*i.e.*, inference) in such models is NP-hard in general [47], early work on structured prediction focused on special cases where the inference task was tractable. Later work

addressed cases where inference was intractable and focused on designing efficient formulations and algorithms.

Existing structured prediction formulations define a score, which is often assumed to consist of multiple local functions, *i.e.*, functions which depend on small subsets of the variables. The parameters of the score function are learned from a given dataset by encouraging that the score for a ground-truth configuration is higher than that of any other configuration. Several works studied the learning problem when inference is hard [13, 26, 40, 17, 33, 23, 43, 35], designing effective approximations.

**Deep Potentials:** After impressive results were demonstrated by Krizhevsky et al. [25] on the ImageNet dataset [12], deep nets gained a significant amount of attention. Alvarez et al. [1], Chen et al. [8], Song et al. [48] took advantage of accurate local classification for tasks such as semantic segmentation by combining deep nets with graphical models in a two step procedure. More specifically, deep nets were first trained to produce local evidence (see Fig. 1a). In a second training step local evidence was fixed and correlations were learned. While leading to impressive results, a two step training procedure seemed counterintuitive and Tompson et al. [53], Zheng et al. [59], Chen* et al. [9], Schwing and Urtasun [42], Lin et al. [29] proposed a unified formulation (see Fig. 1b) which was subsequently shown to perform well on tasks such as semantic segmentation, image tagging *etc*.

Our proposed approach is different in that we combine deep potentials with another deep net that is able to transform the inferred output space or features thereof.

**Autoregressive Models:** Another approach to solve structured prediction problems using deep nets defines an order over the output variables and predicts one variable at a time, conditioned on the previous ones. This approach relies on the chain rule, where the conditional is modeled with a recurrent deep net (RNN). It has achieved impressive results in machine translation [51, 28], computer vision [38], and multi-label classification [36]. The success of these methods ultimately depends on the ability of the neural net to model the conditional distribution, and they are often sensitive to the order in which variables are processed. In contrast, we use a more direct way of modeling structure and a more global approach to inference by predicting all variables together.

**SPENs:** Most related to our approach is the recent work of Belanger and McCallum [3], Belanger et al. [4], which introduced structured prediction energy networks (SPENs) with the goal to address the inductive bias. More specifically, Belanger and McCallum [3] observed that automatically learning the structure of deep nets leads to improved results. To optimize the resulting objective, a relaxation of the discrete output variables to the unit interval was applied and stochastic gradient descent or entropic mirror descent were used. Similar in spirit but more practically oriented is work by Nguyen et al. [37]. These approaches are illustrated in Fig. 1c. Despite additional improvements [4], optimization of the proposed approach remains challenging due to the non-convexity which may cause the output space variables to get stuck in local optima.

'Deep value networks,' proposed by Gygli et al. [15] are another gradient based approach which uses the same architecture and relaxation as SPENs. However, the training objective is inspired by value based reinforcement learning.

Our proposed method differs in two ways. First, we maintain the possibility to explicitly encourage structure inside deep nets. Hence our approach extends SPENs by including additional modeling capabilities. Second, instead of using a continuous relaxation of the output space variables, we formulate inference via a Lagrangian. Due to this formulation we can apply any of the existing inference mechanisms from belief propagation [39] and all its variants [31, 16] to LP relaxations [57]. More importantly, this also allows us (1) to naturally handle problems that are more general than multi-label classification; and (2) to use standard structured loss functions, rather than having to extend them to continuous variables, as SPENs do.

## 3  Model Description

Formally, let $c$ denote input data that is available for conditioning, for example sentences, images, video or volumetric data. Let $x = (x_1, \ldots, x_K) \in \mathcal{X} = \prod_{k=1}^{K} \mathcal{X}_k$ denote the multi-variate output space with $x_k \in \mathcal{X}_k$, $k \in \{1, \ldots, K\}$ indicating a single variable defined on the domain $\mathcal{X}_k \in \{1, \ldots, |\mathcal{X}_k|\}$, assumed to be discrete. Generally, inference amounts to finding the configuration

$x^* = \text{argmax}_{x \in \mathcal{X}} F(x, c, w)$ which maximizes a score $F(x, c, w)$ that depends on the condition $c$, the configuration $x$ and some model parameters $w$.

Classical deep nets assume variables to be independent of each other (given the context). Hence, the score decomposes into a sum of local functions $F(x, c, w) = \sum_{k=1}^{K} f_k(x_k, c, w)$, each depending only on a single $x_k$. Due to this decomposition, inference is easily possible by optimizing each $f_k(x_k, c, w)$ w.r.t. $x_k$ independently of the other ones. Such a model is illustrated in Fig. 1a. It is however immediately apparent that this approach doesn't explicitly take correlations between any pair of variables into account.

To model such context more globally, the score $F(x, c, w) = \sum_{r \in \mathcal{R}} f_r(x_r, c, w)$ is composed of overlapping local functions $f_r(x_r, c, w)$ that are no longer restricted to depend on only a single variable $x_k$. Rather does $f_r$ depend on arbitrary subsets of variables $x_r = (x_k)_{k \in r}$ with $r \subseteq \{1, \ldots, K\}$. The set $\mathcal{R}$ subsumes all subsets $r \in \mathcal{R}$ that are required to describe the score for the considered task. Finding the highest scoring configuration $x^*$ for this type of function generally requires global inference, which is NP-hard [47]. It is common to resort to well-studied approximations [57], unless exact techniques such as dynamic programming or submodular optimization [41, 30, 50, 21] are applicable. The complexity of those approximations increases with the size of the largest variable index subset $r$. Therefore, many of the models considered to date do not exceed pairwise interactions. This is shown in Fig. 1b.

Beyond this restriction to low-order locality, the score function $F(x, c, w) = \sum_{r \in \mathcal{R}} f_r(x_r, c, w)$ being expressed as a sum is itself a limitation. It is this latter restriction which we address directly in this work. However, we emphasize that the employed non-linear output space transformations are able to extract non-local high-order correlations implicitly, hence we address locality indirectly.

To alleviate the restriction of the score function being a sum, and to implicitly enable high-order interactions while modeling structure, our framework extends the aforementioned score via a non-linear transformation of its output, formally,

$$F(x, c, w) = T(c, H(x, c, w), w) . \tag{1}$$

This is illustrated as a general concept in Fig. 1d and with the specific model used by our experiments in Fig. 2. We use $T$ to denote the additional (top) non-linear output transformation. Parameters $w$ may or may not be shared between bottom and top layers, *i.e.*, we view $w$ as a long vector containing all trainable model weights. Different from structured deep nets, where $F$ is required to be real-valued, $H$ may be vector-valued. In this work, $H$ is a vector where each entry represents the score $f_r(x_r, c, w)$ for a given region $r$ and assignment to that region $x_r$, *i.e.*, the vector $H$ has $\sum_{r \in \mathcal{R}} |\mathcal{X}_r|$ entries; however, other forms are possible. It is immediately apparent that[1] $T = \mathbf{1}^\top H$ yields the classical score function $F(x, c, w) = \sum_{r \in \mathcal{R}} f_r(x_r, c, w)$ and other more complex and in particular deep net based transformations $T$ are directly applicable.

Further note that for deep net based transformations $T$, $x$ is no longer part of the outer-most function, making the proposed approach more general than existing methods. Particularly, the 'output space' configuration $x$ is obtained *inside* a deep net, consisting of the bottom part $H$ and the top part $T$. This can be viewed as a *structure-layer*, a natural way to represent meaningful features in the intermediate nodes of the network. Also note that SPENs [3] can be viewed as a special case of our approach (ignoring optimization). Specifically, we obtain the SPEN formulation when $H$ consists of purely local scoring functions, *i.e.*, when $H_k = f_k(x_k, c, w)$. This is illustrated in Fig. 1c.

Generality has implications on inference and learning. Specifically, inference, *i.e.*, solving the program

$$x^* = \underset{x \in \mathcal{X}}{\text{argmax}} \; T(c, H(x, c, w), w) \tag{2}$$

involves back-propagation through the non-linear output transformation $T$. Note that back-propagation through $T$ encodes top-down information into the inferred configuration, while forward propagation through $H$ provides a classical bottom-up signal. Because of the top-down information we say that global structure is implicitly modeled. Alternatively, $T$ can be thought of as an adjustable loss function which matches predicted scores $H$ to data $c$.

Unlike previous structured models, the scoring function presented in Eq. (1) does not decompose across the regions in $\mathcal{R}$. As a result, inference techniques developed previously for structured models

**Algorithm 1** Inference Procedure

---

1: **Input:** Learning rates $\alpha_y$, $\alpha_\lambda$; $y_0$; $\lambda_0$; number of iterations $n$
2: $\mu^* \Leftarrow \operatorname{argmin}_{\hat{\mu}} H^D(\hat{\mu}, c, \lambda, w)$
3: $\bar{\lambda} \Leftarrow \lambda_0$
4: $y_1 \Leftarrow y_0$
5: **for** $i = 1$ **to** $n$ **do**
6:     **repeat**
7:        $y_i \Leftarrow \frac{1}{\alpha_y}\left(y_i - y_{i-1} + \alpha_y \bar{\lambda}\right) - \nabla_y T(c, y, w)$
8:     **until** convergence
9:     $\lambda_i \Leftarrow \lambda_{i-1} - \alpha_\lambda \left(\nabla_\lambda H^D(\mu^*, c, \lambda, w) - y_i\right)$
10:     $\bar{\lambda} = 2\lambda_i - \lambda_{i-1}$; $y_{i+1} \Leftarrow y_i$
11: **end for**
12: $\lambda \Leftarrow \frac{2}{n} \sum_{i=n/2}^{n} \lambda_i$ ; $y \Leftarrow \frac{2}{n} \sum_{i=n/2}^{n} y_i$
13: $\mu \Leftarrow \operatorname{argmin}_{\hat{\mu}} H^D(\hat{\mu}, c, \bar{\lambda}, w)$
14: **Return:** $\mu, \lambda, y$

---

do not apply directly here, and new techniques must be developed. To optimize the program given in Eq. (2), for continuous variables $x$, gradient descent via back-propagation is applicable. In the absence of any other strategy, for discrete $x$, SPENs apply a continuous relaxation where constraints restrict the domain. However, no guarantees are available for this form of optimization, even if maximization over the output space and back-propagation are tractable computationally. Additionally, projection into $\mathcal{X}$ is nontrivial here due to the additional structured constraints. To obtain consistency with existing structured deep net formulations and to maintain applicability of classical inference methods such as dynamic programming and LP relaxations, in the following, we discuss an alternative formulation for both inference and learning.

## 3.1   Inference

We next describe a technique to optimize structured deep nets augmented by non-linear output space transformations. This method is compelling because existing frameworks for graphical models can be deployed. Importantly, optimization over computationally tractable output spaces remains computationally tractable in this formulation. To achieve this goal, a dual-decomposition based Lagrangian technique is used to split the objective into two interacting parts, optimized with an alternating strategy. The resulting inference program is similar in spirit to inference problems derived in other contexts using similar techniques (see, for example, [24]). Formally, note that the inference task considered in Eq. (2) is equivalent to the following constrained program:

$$\max_{x \in \mathcal{X}, y} T(c, y, w) \quad \text{s.t.} \quad y = H(x, c, w), \tag{3}$$

where the variable $y$ may be a vector of scores. By introducing Lagrange multipliers $\lambda$, the proposed objective is reformulated into the following saddle-point problem:

$$\min_{\lambda} \left( \max_{y} \left\{ T(c, y, w) - \lambda^T y \right\} + \max_{x \in \mathcal{X}} \lambda^T H(x, c, w) \right). \tag{4}$$

Two advantages of the resulting program are immediately apparent. Firstly, the objective for the maximization over the output space $\mathcal{X}$, required for the second term in parentheses, decomposes linearly across the regions in $\mathcal{R}$. As a result, this subproblem can be tackled with classical techniques, such as dynamic programming, message passing, *etc.*, for which a great amount of literature is readily available [*e.g.*, 58, 6, 56, 14, 49, 24, 16, 43, 44, 45, 22, 34, 32]. Secondly, maximization over the output space $\mathcal{X}$ is connected to back-propagation only via Lagrange multipliers. Therefore, back-propagation methods can run independently. Here, we optimize over $\mathcal{X}$ by following Hazan et al. [18], Chen$^*$ et al. [9], using a message passing formulation based on an LP relaxation of the original program.

Solving inference requires finding the saddle point of Eq. (4) over $\lambda$, $y$, and $x$. However, the fact that maximization with respect to the output space is a discrete optimization problem complicates this

---

**Algorithm 2** Weight Update Procedure

---

1: **Input:** Learning rate $\alpha$, $\hat{y}$, $\hat{\lambda}$, and $\mathcal{D}$
2: **for** $i = 1$ **to** $n$ **do**
3:     $g = 0$
4:     **for** every datapoint in a minibatch **do**
5:        $\hat{x} \Leftarrow$ Inference in Algorithm 1 (adding $L(x, \hat{x})$)
6:        $g \Leftarrow g + \nabla_w \left( T(c, H(\hat{x}, c, w), w) - T(c, H(x, c, w), w) \right)$
7:     **end for**
8:     $w \Leftarrow w - \alpha \left( Cw + g \right)$
9: **end for**

---

process somewhat. To simplify this, we follow the derivation in [18, 9] by dualizing the LP relaxation problem to convert maximization over $\mathcal{X}$ into a minimization over dual variables $\mu$. This allows us to rewrite inference as follows:

$$\min_\mu \left( \min_\lambda \left( \max_y \left\{ T(c, y, w) - \lambda^T y \right\} + H^D(\mu, c, \lambda, w) \right) \right), \tag{5}$$

where $H^D(\mu, c, \lambda, w)$ is the relaxed dual objective of the original discrete optimization problem. The algorithm is summarized in Alg. 1. See Section 3.3 for discussion of the approach taken to optimize the saddle point.

For arbitrary region decompositions $\mathcal{R}$ and potential transformations $T$, inference can only be guaranteed to converge to local optima of the optimization problem. There do exist choices for $\mathcal{R}$ and $T$, however, where global convergence guarantees can be attained – specifically, if $\mathcal{R}$ forms a tree [39] and if $T$ is concave in $y$ (which can be attained, for example, using an input-convex neural network [2]). We leave exploration of the impact of local versus global inference convergence on model performance for future work. For now, we note that the experimental results presented in Section 4 imply that inference converges sufficiently well in practice for this model to make better predictions than the baselines.

## 3.2 Learning

We formulate the learning task using the common framework for structured support vector machines [52, 54]. Given an arbitrary scoring function $F$, we find optimal weights $w$ by maximizing the margin between the score assigned to the ground-truth configuration and the highest-scoring incorrect configuration:

$$\min_w \sum_{(x,c) \in \mathcal{D}} \underbrace{\max_{\hat{x} \in \mathcal{X}} \left\{ F(\hat{x}, c, w) + L(x, \hat{x}) \right\}}_{\text{Loss augmented inference}} - F(x, c, w) . \tag{6}$$

This formulation applies to any scoring function $F$, and we can therefore substitute in the program given in Eq. (1) to arrive at the final learning objective:

$$\min_w \frac{C}{2} \|w\|_2^2 + \sum_{(x,c) \in \mathcal{D}} \left( \underbrace{\max_{\hat{x} \in \mathcal{X}} \left\{ T(c, H(\hat{x}, c, w), w) + L(x, \hat{x}) \right\}}_{\text{Loss augmented inference}} - T(c, H(x, c, w), w) \right) . \tag{7}$$

To solve loss augmented inference we follow the dual-decomposition based derivation discussed in Sec. 3.1. In short, we replace loss augmented inference with the program obtained in Eq. (5) by adding the loss term $L(x, \hat{x})$. This requires the loss to decompose according to $\mathcal{R}$, which is satisfied by many standard losses (*e.g.*, the Hamming loss). Note that beyond extending SPEN, the proposed learning formulation is less restrictive than SPEN since we don't assume the loss $L(x, \hat{x})$ to be differentiable w.r.t. $x$.

We optimize the program given in Eq. (7) by alternating between inference to update $\lambda$, $y$, and $\mu$ and taking gradient steps in $w$. Note that the specified formulation contains the additional benefit of allowing for the interleaving of optimization w.r.t. $\mu$ and w.r.t. model parameters $w$, though we leave this exploration to future work. Since inference and learning are based on a saddle-point formulation, specific attention has to be paid to ensure convergence to the desired values. We discuss those details subsequently.

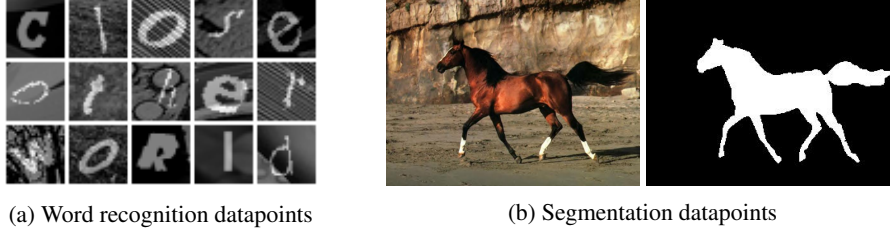

(a) Word recognition datapoints            (b) Segmentation datapoints

Figure 3: Sample datapoints for experiments

Table 1: Results for word recognition experiments. The two numbers per entry represent the word and character accuracies, respectively.

|  | Chain | | | | Second-order | | | |
|---|---|---|---|---|---|---|---|---|
|  | Train | | Test | | Train | | Test | |
| Unary | 0.003 | 0.2946 | 0.000 | 0.2350 | | | | |
| DeepStruct | 0.077 | 0.4548 | 0.040 | 0.3460 | 0.084 | 0.4528 | 0.030 | 0.3220 |
| LinearTop | 0.137 | 0.5308 | **0.085** | 0.4030 | 0.164 | 0.5386 | 0.090 | 0.4090 |
| NLTop | **0.156** | **0.5464** | 0.075 | **0.4150** | **0.242** | **0.5828** | **0.140** | **0.4420** |

## 3.3 Implementation Details

We use the primal-dual algorithm from [7] to solve the saddle point problem in Eq. (5). Though an averaging scheme is not specified, we observe better convergence in practice by averaging over the last $\frac{n}{2}$ iterates of $y$ and $\lambda$. The overall inference procedure is outlined in Alg. 1.

For learning, we select a minibatch of data at every iteration. For all samples in the minibatch we first perform loss augmented inference following Alg. 1 modified by adding the loss. Every round of inference is followed by an update of the weights of the model, which is accomplished via gradient descent. This process is summarized in Alg. 2. Note that the current gradient depends on the model estimates for $\hat{x}$, $\lambda$, and $y$.

We implemented this non-linear structured deep net model using the PyTorch framework.[2] Our implementation allows for the usage of arbitrary higher-order graphical models, and it allows for an arbitrary composition of graphical models within the $H$ vector specified previously. The message passing implementation used to optimize over the discrete space $\mathcal{X}$ is in C++ and integrated into the model code as a python extension.

## 4   Experiments

We evaluate our non-linear structured deep net model on several diverse tasks: word recognition, image tagging, multilabel classification, and semantic segmentation. For these tasks, we trained models using some or all of the following configurations: **Unary** consists of a deep network model containing only unary potentials. **DeepStruct** consists of a deep structured model [9]; unless otherwise specified, these were trained by fixing the pretrained **Unary** potentials and learning pairwise potentials. For all experiments, these potentials have the form $f_{i,j}(x_i, x_j, W) = W_{x_i, x_j}$, where $W_{x_i, x_j}$ is the $(x_i, x_j)$-th element of the weight matrix $W$ and $i,j$ are a pair of nodes in the graph. For the word recognition and segmentation experiments, the pairwise potentials are shared across every pair, and in the others, unique potentials are learned for every pair. These unary and pairwise potentials are then fixed, and a "Top" model is trained using them; **LinearTop** consists of a structured deep net model with linear $T$, *i.e.* $T(c, y, w) = w^T y$, while **NLTop** consists of a structured deep net model where the form of $T$ is task-specific. For all experiments, additional details are discussed in Appendix A.1, including specific architectural details and hyperparameter settings.

**Word Recognition:**   Our first set of experiments were run on a synthetic word recognition dataset. The dataset was constructed by taking a list of 50 common five-letter English words, *e.g.*, 'close,'

Table 2: Results for image tagging experiments. All values are hamming losses.

|          | Train    | Validation | Test   |
|----------|----------|------------|--------|
| Unary    | 1.670    | 2.176      | 2.209  |
| DeepStruct | 1.135  | 2.045      | 2.045  |
| DeepStruct++ | 1.139 | 2.003     | 2.057  |
| SPENInf  | 1.121    | 2.016      | 2.061  |
| NLTop    | **1.111**| **1.976**  | **2.038** |

Table 3: Results for segmentation experiments. All values are mean intersection-over-union

|          | Train    | Validation | Test   |
|----------|----------|------------|--------|
| Unary    | 0.8005   | 0.7266     | 0.7100 |
| DeepStruct | 0.8216 | 0.7334     | 0.7219 |
| SPENInf  | **0.8585** | **0.7542** | **0.7525** |
| NLTop    | **0.8542** | **0.7552** | **0.7522** |
| Oracle   | 0.9260   | 0.8792     | 0.8633 |

'other,' and 'world,' and rendering each letter as a 28x28 pixel image. This was done by selecting a random image of each letter from the Chars74K dataset [11], randomly rotating, shifting, and scaling them, and then inserting them into random background patches with high intensity variance. The task is then to identify each word from the five letter images. The training, validation, and test sets for these experiments consist of 1,000, 200, and 200 words, respectively, generated in this way. See Fig. 3a for sample words from this dataset.

Here, **Unary** consists of a two-layer perceptron trained using a max-margin loss on the individual letter images as a 26-way letter classifier. Both **LinearTop** and **NLTop** models were trained for this task, the latter of which consist of 2-layer sigmoidal multilayer perceptrons. For all structured models, two different graphs were used: each contains five nodes, one per letter in each word. The first contains four pair edges connecting each adjacent letter, and the second additionally contains second-order edges connecting letters to letters two positions away. Both graph configurations of the LinearTop and NLTop models finished 400 epochs of training in approximately 2 hours.

The word and character accuracy results for these experiments are presented in Tab. 1. We observe that, for both graph types, adding structure improves model performance. Additionally, including a global potential transformation increases performance further, and this improvement is increased when the transformation is nonlinear.

**Multilabel Classification:** For this set of experiments, we compare against SPENs on the Bibtex and Bookmarks datasets used by Belanger and McCallum [3] and Tu and Gimpel [55]. These datasets consist of binary feature vectors, each of which is assigned some subset of 159/208 possible labels, respectively. 500/1000 pairs were chosen for the structured models for Bibtex and Bookmarks, respectively, by selecting the labels appearing most frequently together within the training data.

Our **Unary** model obtained macro-averaged F1 scores of 44.0 and 38.4 on the Bibtex and Bookmarks datasets, respectively; **DeepStruct** and **NLStruct** performed comparably. Note that these classifier scores outperform the SPEN results reported in Tu and Gimpel [55] of 42.4 and 34.4, respectively.

**Image Tagging:** Next, we train image tagging models using the MIRFLICKR25k dataset [20]. It consists of 25,000 images taken from Flickr, each of which are assigned some subset of a possible 24 tags. The train/development/test sets for these experiments consisted of 10,000/5,000/10,000 images, respectively.

Here, the **Unary** classifier consists of AlexNet [25], first pre-trained on ImageNet and then fine-tuned on the MIRFLICKR25k data. For **DeepStruct**, both the unary and pairwise potentials were trained jointly. A fully connected pairwise graphical model was used, with one binary node per label and an edge connecting every pair of labels. Training of the **NLStruct** model was completed in approximately 9.2 hours.

The results for this set of experiments are presented in Tab. 2. We observe that adding explicit structure improves a non-structured model and that adding implicit structure through $T$ improves an explicitly structured model. We additionally compare against a SPEN-like inference procedure (**SPENInf**) as follows: we load the trained **NLTop** model and find the optimal output structure $\max_{x \in \mathcal{X}} T(c, H(c, w, w), w)$ by relaxing $x$ to be in $[0, 1]^{24}$ and using gradient ascent (the final output is obtained by rounding). We observe that using this inference procedure provides inferior results to our approach.

To verify that the improved results for NLTop are not the result of an increased number of parameters, we additionally trained another **DeepStruct** model containing more parameters, which is called **DeepStruct++** in Table 2. Specifically, we fixed the original **DeepStruct** potentials and learned two additional 2-layer multilayer perceptrons that further transformed the unary and pairwise potentials. Note that this model adds approximately 1.8 times more parameters than **NLTop** (*i.e.*, 2,444,544 *vs.* 1,329,408) but performs worse. NLTop can capture global structure that may be present in the data during inference, whereas **DeepStruct** only captures local structure duing inference.

**Semantic Segmentation:** Finally, we run foreground-background segmentation on the Weizmann Horses database [5], consisting of 328 images of horses paired with segmentation masks (see Fig. 3b for example images). We use train/validation/test splits of 196/66/66 images, respectively. Additionally, we scale the input images such that the smaller dimension is 224 pixels long and take a center crop of 224x224 pixels; the same is done for the masks, except using a length of 64 pixels. The **Unary** classifier is similar to FCN-AlexNet from [46], while **NLStruct** consists of a convolutional architecture built from residual blocks [19]. We additionally train a model with similar architecture to **NLStruct** where ground-truth labels are included as an input into $T$ (**Oracle**). Here, the **NLStruct** model required approximately 10 hours to complete 160 training epochs.

Tab. 3 displays the results for this experiment. Once again, we observe that adding the potential transformation $T$ is able to improve task performance. The far superior performance by the **Oracle** model validates our approach, as it suggests that our model formulation has the capacity to take a fixed set of potentials and rebalance them in such a way that performs better than using those potentials alone. We also evaluate the model using the same SPEN-like inference procedure as described in the the Image Tagging experiment (**SPENInf**). In this case, both approaches performed comparably.

## 5  Conclusion and Future Work

In this work we developed a framework for deep structured models which allows for implicit modeling of higher-order structure as an intermediate layer in the deep net. We showed that our approach generalizes existing models such as structured prediction energy networks. We also discussed an optimization framework which retains applicability of existing inference engines such as dynamic programming or LP relaxations. Our approach was shown to improve performance on a variety of tasks over a base set of potentials.

Moving forward, we will continue to develop this framework by investigating other possible architectures for the top network $T$ and investigating other methods of solving inference. Additionally, we hope to assess this framework's applicability on other tasks. In particular, the tasks chosen for experimentation here contained fixed-size output structures; however, it is common for the outputs for structured prediction tasks to be of variable size. This requires different architectures for $T$ than the ones considered here.

**Acknowledgments:** This material is based upon work supported in part by the National Science Foundation under Grant No. 1718221, Samsung, 3M, and the IBM-ILLINOIS Center for Cognitive Computing Systems Research (C3SR). We thank NVIDIA for providing the GPUs used for this research.

## Footnotes

[1] $\mathbf{1}$ denotes the all ones vector.

[2]Code available at: `https://github.com/cgraber/NLStruct`.

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
