[Supplementary Material]

# A   Appendix

## A.1   Experimental Details

For all experiments, learning rate, weight decay coefficient $C$, and number of epochs were tuned using a validation set of data; for datasets without a specified validation dataset, a portion of the training data was held-out and used instead to tune parameters. All models were trained using minibatch gradient descent. Additionally, all structured models (everything besides the **Unary** configuration) used loss-augmentation terms $L(x, \hat{x}) = \mathbf{1}[x \neq \hat{x}]$ during training. For **NLTop** models, the number of iterations per round of inference $n$ was set to 100; this value was chosen based on preliminary experimentation.

**Word recognition experiments**   The **Unary** classifier model consists of a two-layer multilayer perceptron with 128 hidden units and a ReLU nonlinearity. The multilayer perceptron used in the **NLTop** configuration contains 2834 hidden units, which is equal to the size of the $y$ and $\lambda$ vectors, and used sigmoid activation functions. The weights for the first layer of this model were initialized to the identity matrix, and the weights for the second layer were initialized to a vector of all 1s.

**Multilabel classification experiments**   The **Unary** classifier model consists of two-layer multilayer perceptrons with ReLU nonlinearities and 318/600 hidden units for Bibtex/Bookmarks, respectively. For all pairs of nodes $i$ and $j$, pairwise potentials were constrained such that $W_{0,0} = W_{1,1}$ and $W_{1,0} = W_{0,1}$. For the Bibtex experiment, the **NLTop** $T$ model is a 2-layer mulitlayer perceptron with 1000 hidden units and uses leaky ReLU activation functions with negative slopes of 0.25. For the Bookmarks experiment, the **NLTop** $T$ model is a 2-layer multilayer perceptron with 4000 hidden units and sigmoid activation functions; the potentials were divided by 100 before being input into $T$.

**Image tagging experiments**   The **Unary** classifier model consists of the pre-trained Alexnet model provided by PyTorch, with the final classifier layer stripped and replaced to generate unary potentials. For all pairs of nodes $i$ and $j$, pairwise potentials were constrained such that $W_{0,0} = W_{1,1}$ and $W_{1,0} = W_{0,1}$. The multilayer perceptron used in the **NLTop** configuration contains 1152 hidden units and used hardtanh activation functions. For the configuration with additional parameters, the additional MLPs contained hidden sizes equal to both the input and output sizes, which were equal to the number of unary/pairwise potentials, respectively. Each MLP took one of these sets of potentials as input and transformed them, leading to a new set of potentials. For the **SPENInf** experiments, we run inference using 5 random initializations of $x$ and report the average task losses. The standard deviations for Train, Validation, and Test sets were 0.0012, 0.0008, and 0.0007, respectively.

**Semantic segmentation experiments**   **Unary** consists of an AlexNet model [25] pretrained on ImageNet with the first MaxPool layer removed and the stride of the second MaxPool Layer from 2 to 1. Additionally, all fully connected layers are replaced with 1x1 convolutional layers, and the final classifier layer is replaced with a 1x1 convolutional layer that outputs two channels, one for each of the two possible labels. Finallly, a deconvolutional layer with a kernel of size 14x14 is used to upsample the output of the previous part to the final 2x64x64 potential maps. The architecture used for **NLTop** is presented in Fig. 4. The input to $T$ contains 13 channels of information: two of these channels consist of the unary potentials, with one channel per possible output value; four of these channels consist of the row pairwise potentials, with one channel per possible pair of output values (and one column of padding, since there are $n - 1$ pairs along a row with $n$ nodes); four of these channels consist of the column pairwise potentials, with one channel present per possible pair of output values (and one row of padding); and three of these channels consist of the (normalized) input images). The **Oracle** experiments use the same architecture, except with 23 channels of information being used (the additional ten channels being the ground-truth beliefs, reshaped in the same manner as the potentials). Hence, for this experiment, every convolutional layer except the last contains 23 filters.

Figure 4: The $T$ model used for Semantic Segmentation experiments.