[Reviews · NeurIPS 2018]

Reviewer 1



This paper studies the problem of training deep structured models (models where the dependencies between the output variables are explicitly modelled and some components are modelled via neural networks). The key idea of this paper is to give up the standard modelling assumption of structured prediction: the score (or the energy) function is the sum of summands (potentials). Instead of using the sum the paper puts an arbitrary non-linear (a neural network) transformation on top of the potentials. The paper develops an inference (MAP prediction) technique for such models which is based on Lagrangian decomposition (often referred to as dual decomposition, see details below). The training of the model is done by combining this inference technique with the standard Structure SVM (SSVM) objective. The proposed approach is evaluated in 4 settings: OCR, multilabel classification, image tagging, sematic segmentation and is compared against several baselines. Overall the paper is well written (I think I understood the method without too much effort) and there is a novel component. However, I don't think that the paper is significant enough to warrant a NIPS publication. My skepticism is based on the following three points: 1) The proposed method has no theoretical justification on the inference side. The Lagrangian relaxation technique usually works well when the duality gap is small (Kappes et al., 2015), but can provide arbitrarily bad results otherwise. There is no study of whether the inference problem is solved well or not. 2) The paper uses non-standard datasets and does not compare to any baselines reported in the literature. For OCR, there is a standard dataset of Taskar et al. (2003) and a lot of baselines (see, "Lacoste-Julien et al., Block-Coordinate Frank-Wolfe Optimization for Structural SVMs, ICML 2013" for linear baselines; "Perez-Cruz, Ghahramani, Pontil, Conditional Graphical Models, 2006, http://mlg.eng.cam.ac.uk/zoubin/papers/CGM.pdf" for kernel-based baselines, "Leblond et al., SEARNN: Training RNNs with Global-Local Losses, 2017, https://arxiv.org/abs/1706.04499" for some RNN-based baselines). I do not understand the reasoning for not using the standard dataset on the same tasks. 3) The novelty of the paper is somewhat limited, in part because learning with SSVM and Lagrangian dual of the inference procedure was done before. In particular, the work of Komodakis (2011) did this, but connections to it are not discussed at all (the paper is cited however). The work of Komodakis&Paragios (2009) used an inference method, which is very similar to the one proposed, and is directly comparable in some settings. For example, the OCR datasets (both the version used for experiments and the standard one of Taskar et al. (2003)) contain only 50 different words, this ideally matches the patter-based potentials used by Komodakis&Paragios (2009). I do agree that the proposed method is not identical to the one of Komodakis&Paragios (2009), but I would insist on fair discussion and comparison. === after the response I would like to thank the authors for addressing some of my questions and hope the corresponding clarifications would be added to the next revision. However, I still think that the contribution is somewhat incremental because the setting is quite limited, but anyway I'm raising my score to 6.

Reviewer 2



This work proposes an approach to model the output structure inside the neural network to be optimized and treats the prediction output as an "intermediate" layer, which is further transformed in order to model any global interactions between sub-parts of the structured output. This paper is heavily motivated by the idea behind Structured Prediction Energy Networks(SPENs), but is different mainly in implementation. While SPENs rely on making the output space continuous in order to be able to differentiate wrt output to estimate an approximate maximum scoring output under current parametrization and then define a loss on this estimate for estimating the parameters via backprop; this paper in contrast proposes a lagrangian relaxation (and a further LP relaxation) based procedure that poses loss augmented inference as a relaxed discrete optimization and uses it to compute a margin loss that is used for training the weights of the network. They perform experiments on three tasks comparing to "Unary" models and "DeepStruct" (fixed unary + learned pairwise) models and showing improvements. My main concern is that more details are required for differentiability of the inference procedure. Going from eq 4 to 5 for example involves a relaxation of a discrete objective, which makes it very similar to SPENs as they rely on continuous relaxations as well. Is \hat{x} explicitly identified? How is it identified from the dual variables \mu, \lambda, y yielded by algorithm 1? It seems like \hat{x} in the primal problem is important so that the loss L(x,\hat{x}) can be computed as well. How are you incorporating the loss into the inference procedure? Are their any assumptions on the loss function for tractable/practical inference? I couldn't find such details in the paper. This detail is also important because it is important to identify if there are any discontinuous operations in the inference procedure which would not be amenable for backpropagation. SPENs pay careful attention to the differentiability of the whole procedure for end-to-end backpropagation and other work (Goyal et al.,2018) on beam search aware training of neural models also proposes continuous relaxation of the beam search procedure to make end-to-end backpropagation possible. With a discontinuous inference procedure, the optimization of neural network parameters would be blind to training signal from the inference procedure. From the text in the paper, I am not sure how this issue is being addressed in the proposed approach. Also, I couldn't find "multilabel classification" numbers in the paper. Am I missing something? For DeepStruct model, why are the pretrained unary potentials fixed? Previous work has shown that finetuning (or further training wrt the unary potentials) is helpful. This is important because this might lead to a stronger baseline to compare against. Overall, I liked the motivation behind the problem and the proposed approach and experiemnts seem sound. However, some important details are missing for a more thorough assessment of the paper and the baselines could be stronger.

Reviewer 3



Main idea: This paper extends SPEN by considering a special architecture of the energy network, where the potential/energy vector is explicitly computed as a hidden layer. This enables a dual decomposition inference algorithm, thus somehow circumvents the discrete optimization over the structured output space, which was the main difficulty/restriction of SPEN. Strength: I would like to mention that the idea is a natural extension of structured SVM with dual decomposition inference [*, **] to SPEN. It would be better if the connection was discussed at Sec 3.1. I really like the way dual decomposition applies. It doesn't decompose over factors as originally it was intended to, but it separates the discrete optimization out from the continuous one, which sheds light on how we deal with discrete variables in a neural network. Clarity: - A better explanation of Fig 1 should be added. It is really hard to see why SPEN corresponds "independent" while the proposed method is labeled structured. - In (1) and L153, the difference between SPEN and the proposed T(c,H(x,c)) is not clear. I don't understand why is this more general than SPEN given the definition of SPEN is quite general as a neural network taking both x and c as the input. Weakness: - (4) or (5) are nonconvex saddle point problems, there is no convergence guarantee for Alg 1. Moreover, as a subroutine for (7), it is not clear how many iterations (the hyperparameter n) should be taken to make sure (7) is convergent. Previously in structured SVM, people noticed that approximate inference could make the learning diverges. - Performance boost due to more parameters? In Tab 1,2,3, if we think carefully, LinearTop and NLTop adds additional parameters, while Unary performs much worse comparing to the numbers reported e.g. in [14], where they used a different and probably better neural network. This raises a question: if we use a better Unary baseline, is there still a performance boost? - In Table 1, the accuracies are extremely poor: testing accuracy = 0.0? Something must be wrong in this experiment. - Scalability: since H(x,c) outputs the whole potential vector with length O(K^m), where m is the cardinality of the largest factor, which could be extremely long to be an input for T. - The performance of NLTop is way behind the Oracle (which uses GT as input for T). Does this indicate (3) is poorly solved or because of the learning itself? [*] N Komodakis Efficient training for pairwise or higher order CRFs via dual decompositio. CVPR 2011. [**] D Sontag et al. Learning efficiently with approximate inference via dual losses. ICML 2010.